# How to Select Slices for Annotation to Train Best-performing Deep Learning Segmentation Models for Cross-sectional Medical Images?

**Yixin Zhang**[1]  iD                                                    YIXIN.ZHANG7@DUKE.EDU
[1] *Department of Electrical and Computer Engineering, Duke University, NC, USA*
**Kevin Kramer**[2]                                                          KEVIN@MINNHEALTH.COM
[2] *Minnesota Health Solutions, Minneapolis, MN 55414, USA*

**Maciej A. Mazurowski**[1,3,4,5]                                    MACIEJ.MAZUROWSKI@DUKE.EDU
[3] *Department of Radiology, Duke University, NC, USA*

[4] *Department of Biostatistics & Bioinformatics, Duke University, NC, USA*

[5] *Department of Computer Science, Duke University, NC, USA*

**Editors:** Accepted for publication at MIDL 2025

## Abstract

Automated segmentation of medical images heavily relies on the availability of precise manual annotations. However, generating these annotations is often time-consuming, expensive, and sometimes requires specialized expertise (especially for cross-sectional medical images). Therefore, it is essential to optimize the use of annotation resources to ensure efficiency and effectiveness. In this paper, we systematically address the question: "in a non-interactive annotation pipeline, how should slices from cross-sectional medical images be selected for annotation to maximize the performance of the resulting deep learning segmentation models?" We conducted experiments on 4 medical imaging segmentation tasks with varying annotation budgets, numbers of annotated cases, numbers of annotated slices per volume, slice selection techniques, and mask interpolations. We found that: 1) It is almost always preferable to annotate fewer slices per volume and more volumes given an annotation budget. 2) Selecting slices for annotation by unsupervised active learning (UAL) is not superior to selecting slices randomly or at fixed intervals, provided that each volume is allocated the same number of annotated slices. 3) Interpolating masks between annotated slices rarely enhances model performance, with exceptions of some specific configuration for 3D models.

**Keywords:** Annotation, Semantic segmentation, Cross-sectional Medical Image Analysis

## 1. Introduction

Deep learning models based on Convolutional Neural Networks (CNNs) and Visual Transformers (ViTs) have achieved remarkable success across a wide range of computer vision tasks. However, such success relies heavily on large, annotated datasets, which are often scarce in medical imaging. While fine-tuning models pre-trained on abundant natural image datasets is a conceptually appealing solution, its effectiveness is limited due to significant discrepancies in the feature spaces of natural and medical images. (Alzubaidi et al., 2021). As a result, researchers still often need to create new annotations for their specific medical image segmentation tasks. Due to budget constraints, it is often impractical for researchers to annotate all slices within every available 3D volume. A common alternative strategy is

to create sparse annotations, where only a subset of (ideally representative) slices is annotated, thereby reducing the workload on annotators. This approach is related to weakly supervised learning (WSSS) (Vorontsov and Kadoury, 2021; Zlateski et al., 2018; Zhang et al., 2023) in that only partial information is provided to the model. However, it differs in that sparse annotations involve pixel-wise precise labels on the annotated slices, as opposed to the coarse or incomplete labels typically used in WSSS. Çiçek et al. found that 3D-UNet trained on sparse annotations can perform comparably to those trained on dense annotations for kidney segmentation. (Çiçek et al., 2016) Still, whether the merits of sparse annotation holds across diverse types of objects is untested. Plus, the optimal number of volumes and slices per volume, and which slices should be annotated remains unclear.

In parallel with the study on the impact of various annotation configurations on different target types, we also evaluate a complementary approach for utilizing sparse annotation known as mask interpolation (M.I.), which leverages sparse annotations to generate dense annotations in a self-supervised manner before training. Yeung et al. introduced Sli2Vol, a self-supervised framework that reconstructs annotations for unlabeled slices based on a reference slice, converting sparsely annotated volumes into densely annotated ones with a combination of human-annotated and algorithm-generated masks. (Yeung et al., 2021) Wu et al. proposed Single Slice Annotation (SSA) with similar purpose but incorporated more sophisticated pre-processing and regularization during model training for better robustness. (Wu et al., 2022)

## 2. Methods

Our study is developed along three aspects on how one could improve the cost-effectiveness in their slice selection for sparse annotation. For the simplicity and uniformity of experiment design, we assume that within each dataset, 1) annotations are performed on slices aligned along the axial axis (Superior-Inferior direction); 2) Supervised training begins only after the full set of annotations is complete; 3) All voxels in the dataset have consistent physical spacing; 4) An equal number of slices are chosen from each volume for annotation. 5) The cost for annotating each batch of N nearly unbiasedly selected slices for the same class of target is approximately consistent.

### 2.1. Fewer Annotated Slices Per Volume for More Volumes or More Annotated Slices Per Volume for Less Volumes?

Our primary experiment explore whether it is more advantageous to annotate a few volumes with more annotated slices per volume or to spread annotations across a larger number of volumes with fewer annotated slices per volume. We denote the two parameters involved in this experiment as annotation density ($\rho$) and volume count (s), respectively. Annotation density ($\rho$) represents the percentage of annotated slices per volume. It takes values from the set $\{5\%, 10\%, 20\%, 40\%, 80\%\}$. Volume count (s) represents the fraction of volumes with annotations from all volumes reserved in the original training set. It takes value from the set $\{1/8, 1/4, 1/2, 1/1\}$. For a dataset configuration with ($\rho = 10\%$, s $= 1/4$), one-fourth of all available training volumes are selected for this dataset, and 10% slices in each selected volume are annotated. Multiplying annotation density and volume count will then give the total annotation budget as the fraction of the budget for all volumes and all slices.

For each dataset, across different combinations of parameters $(\rho, s)$, we generate sparse annotations from the corresponding dense annotations using the three slice selection methods described in Section 4.3. Each configuration is used to train models three times, resulting in 9 trials per $(\rho, s)$ pair for each dataset. We evaluate the resultant models' performance using the average batched Intersection over Union (IoU), and report the mean and standard deviation in Tables 2 and 3 under Appendix C. A reader-friendly visualization of the results is provided in Section 4.1.

## 2.2. Does Mask Interpolation (M.I.) help?

In parallel with analyzing the trade-off between $\rho$ and $s$, we investigated whether the complementary practice of applying mask interpolation (M.I.) to preprocess sparse annotations into noisy dense ones benefits the training of both 2D and 3D models. To begin, we verified that our implementation of M.I. achieved performance comparable to Sli2Vol (Yeung et al., 2021) and SSA (Wu et al., 2022) on organ-like objects. We describe our implementation, and training pipeline M.I. in Appendix F, the procedure for generating pseudo-masks for the unannotated slices from a sparsely annotated dataset in Algorithm 1, and presents the quality (measured by IoU) of the interpolated annotation in Table 5 under Appendix E.

With the annotation budget distributed across all available volumes (i.e., $s = 1$), we vary the annotation density $\rho$ to assess whether, and under what conditions, models trained with mask interpolation (M.I.) outperform those where the gradients of unannotated slices or voxels are eliminated. The mean and standard deviation of the results are reported in Table 4 in Appendix D, and a reader-friendly visualization is provided in Section 4.2.

## 2.3. Impact of Slice Selection Methods within Volumes

Assuming a fixed configuration of annotation density $(\rho)$, volume count (s), we also empirically compared the efficacy of three slice selection methods for annotation. Two of the methods, Selection at fixed interval (i.e., annotating every n-th slice) and Selection at uniform random (i.e., randomly picking slices with uniform distribution), are simple yet effective for unbiased slice selection with a wide coverage of locations within each volume. A third approach to select slice leverages the concept of Active Learning (AL) (Ren et al., 2021; Gal et al., 2017; Zhang et al., 2019), which seeks to reduces labeling costs by focusing on the most informative samples. However, unlike traditional AL methods that require human-computer interaction and the availability of some advanced data engines, we focus on a specific variant of AL based on Unsupervised Active Learning (UAL). This class of methods claim to preserve the key advantage of selecting the "most informative or representative subset of samples" within a fixed annotation budget, while eliminating the need for human interaction with the model during the slice selection process. In this study, our implementation of UAL is based on Representative Annotation (Zheng et al., 2019), which laid the foundation for many subsequent unsupervised active learning (UAL) methods. In our implementation, we first trained a Masked Autoencoder (MAE) for feature extraction and then utilized these features to compute the representativeness score. This approach also aligns with the core-set optimization strategy (Sener and Savarese, 2018).

For each $(\rho, s)$ configuration, we generate the corresponding sparse annotation for each dataset using each of the aforementioned slice selection method and conduct three trials of

model training on these sparsely annotated dataset. We then record the disparity between the average performance of each slice selection method and the overall average performance across all trials. The results are presented in Figure 5 as a box plot. If a slice selection method is superior, its bar should be noticeably higher than those of the other methods for the same dataset.

## 3. Datasets and Model Architectures

Our experiments involved three datasets across four tasks. Table 1 lists the dataset specifications after preprocessing, with the primary target of interest being (1) liver tumor for LiTS17; (2) breast and (3) fibroglandular tissue, abbreviated as FGT, for DBC; and (4) trace of stroke for ATLAS. We selected representative segmentation models and paired them with each dataset and tuned hyper-parameters to ensure that models trained on dense annotations yield comparable performance to existing literature on similar tasks. We attach a visualization for each type of target in Appendix A, the rational of choosing these datasets in Appendix B, and the model training details in Appendix G.

| Dataset/Target | Spacing (mm) | Dimensions | #Train | #Val | #Test |
|---|---|---|---|---|---|
| LiTS17 (Bilic et al., 2023) | (1, 1, 1) | 384×384×384 | 96 | 10 | 25 |
| DBC (Breast)(Saha et al., 2018) | (0.7, 0.7, 1) | 496×496×176 | 72 | 8 | 20 |
| DBC (FGT) (Saha et al., 2018) | (0.7, 0.7, 1) | 496×496×176 | 72 | 8 | 20 |
| ATLAS (Liew et al., 2022) | (1, 1, 1) | 224×256×189 | 500 | 55 | 100 |

Table 1: Specifications of the 3D volumes after preprocessing for each dataset/target.

## 4. Results

### 4.1. Fewer Annotated Slices Per Volume for More Volumes or Converse?

#### 4.1.1. 2D Segmentation Setting

We present in Figure 1 the average drop in model performance across the dataset after replacing the dense annotation to sparse ones. In 2D models, the number of annotated volumes appears to be a more significant factor influencing the performance of sparsely annotated datasets. Distributing the annotation budget across a greater number of volumes, even with lower annotation density, consistently results in models that exhibit a smaller performance gap compared to those trained on densely annotated datasets.

For LiTS17 (Liver CT, Tumor, Fig. 2, first column to the left), higher annotation density ($\rho$) consistently resulted in lower performance when compared to increasing volume counts, regardless of the budget. Doubling $\rho$ did not improve performance as effectively as doubling the volume count (s). The same performance trend observed in LiTS17 was seen in both tasks of the DBC datasets (Breast MRI, Breast and Fibroglandular Tissue, Fig. 2, second and third column to the left). For these tasks, improving performance was more effectively achieved by increasing the volume count rather than the annotation density. In the ATLAS dataset (Brain MRI, Trace of Stroke, Fig. 2, forth column to the left), we observed a unique outcome at four budget levels (1181, 2362, 4725, and 37800 slices): some lower volume count

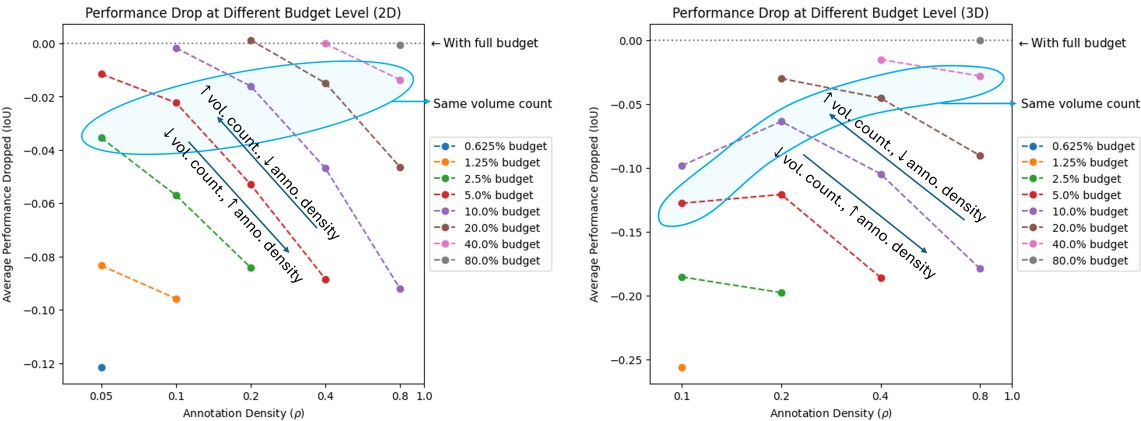

Figure 1: Average model performance drops when sparsely annotated datasets with different annotation density and volume counts are used for training 2D (left) and 3D (right) models. Volume counts are adjusted to match the annotation budget.

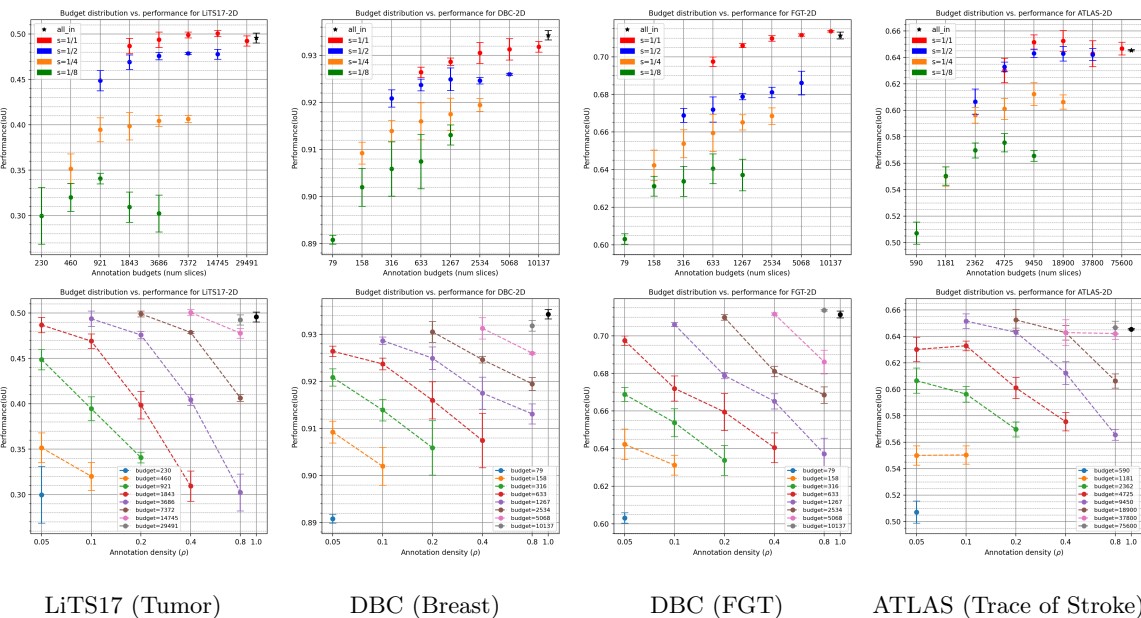

LiTS17 (Tumor)    DBC (Breast)    DBC (FGT)    ATLAS (Trace of Stroke)

Figure 2: First row: Performance (y-axis, by batched IoU) of 2D models when trained on sparsely annotated datasets with different configurations of annotation budget (x-axis) and volume count (by color). Second Row: Performance of 2D models trained with different (annotation density, volume count) configurations. Points on the same line (same color) consume the same budgets (i.e., annotated slices).

configurations achieved comparable or even superior performance. There are two cases for such exception, one occurs when both the total budget and annotation density are small

(i.e., budget $\leq$ 4725 slices, $\rho \leq 0.1$). The other occurs when both the total budget and annotation density are large (i.e., budget $\geq$ 37800 slices, $\rho \geq 0.4$).

Based on these results, if a dataset is prepared for training 2D models, spreading annotation budgets across more volumes at diverse locations is always more cost-effective. A small annotation density (e.g., 5% ~ 20% slices labeled) is often sufficient for training models with similar performance to the counterparts trained with the full dataset.

### 4.1.2. 3D Segmentation Setting

In the 3D segmentation setting, most patterns observed in the 2D models also apply, but with some notable differences. As shown in Fig. 1 (right), we observed that excessively low annotation density can hinder model performance in certain conditions. For all tasks

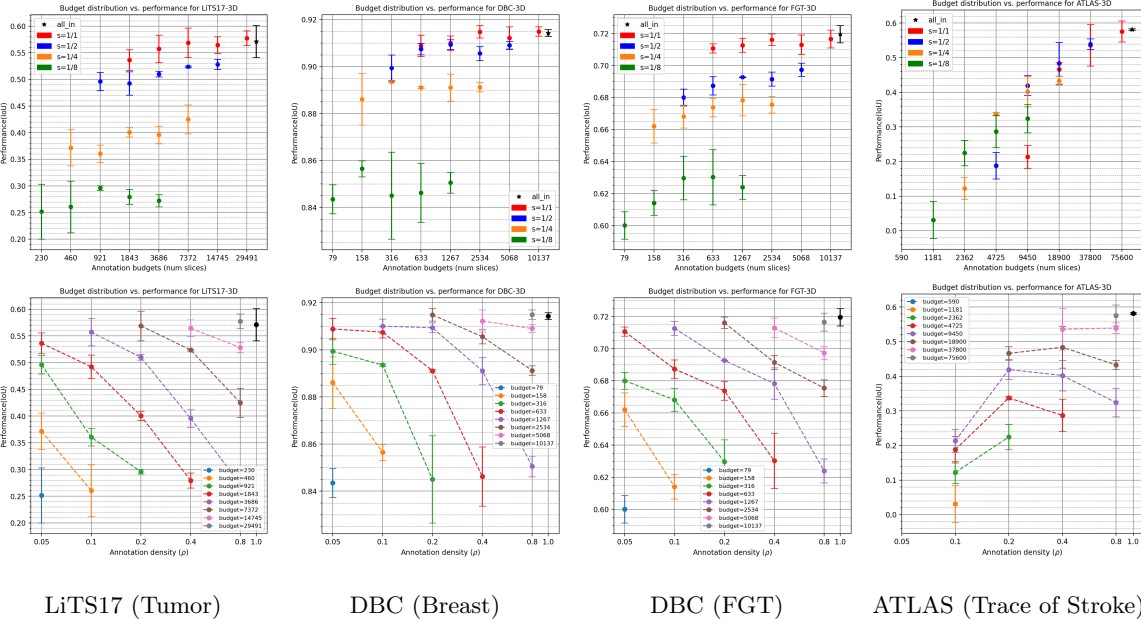

Figure 3: First row: Performance (y-axis, by IoU) of 3D models when trained on sparsely annotated datasets with different configurations of annotation budget (x-axis) and volume count (by color). Second Row: Performance of 3D models trained with different (annotation density, volume count) configurations. Points on the same line (same color) consume the same budgets (i.e., annotated slices).

of LiTS17 and DBC in 3D, their trends mirror those in 2D (Fig. 3, first three columns to the left): low annotation density does not significantly affect performance due to clear object-background contrast. However, for the ATLAS dataset, the behavior was slightly different. At low annotation density $\rho = 0.05$, the model failed to learn useful features and showed no prediction capability, thus omitted in our report. Instead, increasing annotation density in the range of $0.1 \leq \rho \leq 0.4$ led to noticeable performance improvements, as shown in (Fig. 3, forth column to the left). We will expand on this observation in Section 6.

## 4.2. Should Mask Interpolation (M.I.) Be Used?

We examined whether imputing labels for unannotated voxels before training provides an advantage over training directly with sparse annotations.

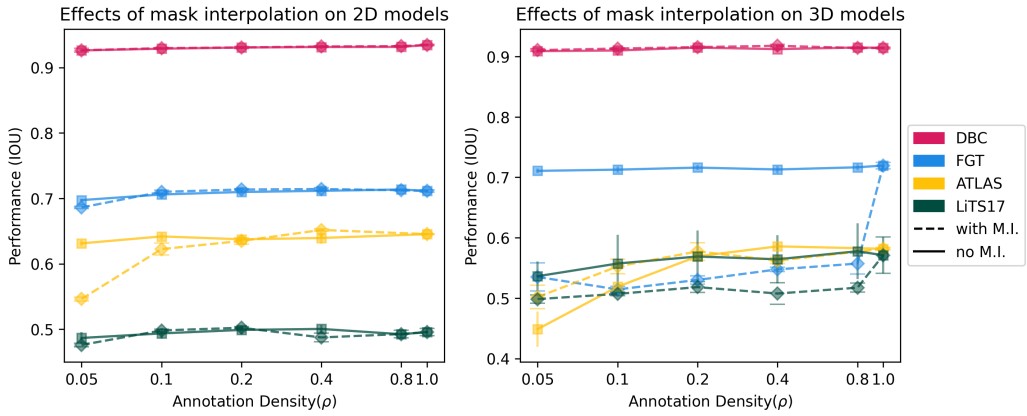

Figure 4: Performance of 2D and 3D models when trained on the same set of sparsely annotated volumes before and after the use of mask interpolation. The horizontal and vertical line segments plotted around the data points indicates the standard deviation for the "with M.I." and "no M.I." variants respectively.

According to Figure 4, applying mask interpolation before model training showed no improvement in the performance of 2D models on any dataset. In 3D segmentation, the effects varied: for DBC breast segmentation, the impact was minimal, while LiTS17 tumor segmentation often experienced a minor performance drop. A statistically significant performance drops were noted with FGT (fibro-glandular tissue segmentation). Conversely, ATLAS showed improved performance at low annotation densities ($\rho \leq 0.2$).

## 4.3. Is the way slices are selected for annotation important?

We abbreviate the three slice selection methods described in Section 2.3 as (rand, fixed, UAL) for 1) Selection at fixed interval; 2) Selection at uniform random; and 3) UAL-based selection. Now, we examine the impacts of the different slice selection methods. Fig. 5

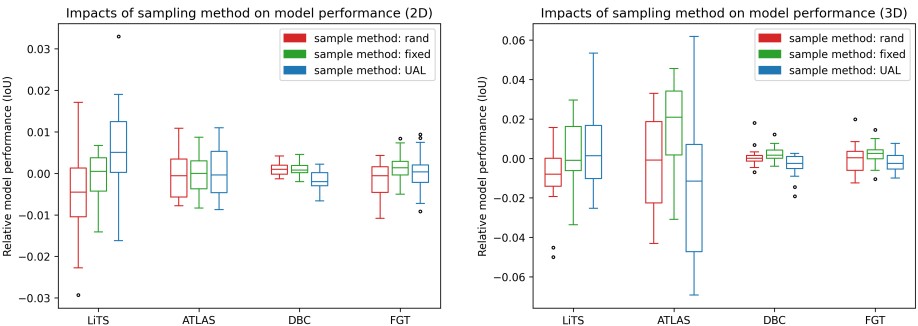

Figure 5: Distribution of relative performance for different slices selection methods

shows that the performance differences between the three slice selection methods are minor. Neither method consistently outperforms the others in 2D or 3D settings. In 3D models, while overall trends are similar to those in 2D, there is greater variation in how different selection policies affect performance. We discuss these observations in Section 6.

## 5. Limitation

In this study, we acknowledge the following aspects to have potential limitations:

- Our study did not include HD95 as a performance metric.

- Due to computational constraints, we applied Unsupervised Active Learning (UAL) at the volume level rather than across the entire dataset. However, our results suggest that for UAL to be effective, it must be deployed at the dataset level rather than within individual volumes. Future studies on other ways of applying AL is needed.

- We evaluated our approach on three datasets, which, while diverse, may not fully capture the full spectrum of medical imaging applications. Expanding to additional datasets and tasks could further validate the generalizability of our findings.

- Further research is needed to investigate the interaction between datasets and architectures to better understand the phenomenon observed with ATLAS under the 3D segmentation setting.

- Our study assumes each nearly unbiasedly sampled batch of slices to consume approximately consistent time and resources to annotate. However, this assumption may not hold with higher annotation density, as multiple slices are likely to capture the sample object, reducing the time needed for the annotation to localized the object to annotate. Further study on the modeling of such change in time ratio with be beneficial in refining our conclusion.

## 6. Discussion and Conclusion

In our 2D segmentation experiments, we found it advantageous to distribute the annotation budget across a greater number of volumes with fewer annotated slices per volume. This approach helps mitigate redundancy in adjacent slices, as focusing annotation on fewer volumes results in many highly similar slices. Annotating a broader range of volumes introduces the model to a wider variety of naturally-occurred variations between volumes, which data augmentation may not replicate. In 3D deep learning methods, the principle that "sparser annotations are better" generally holds true as well, despite the exception we noticed on the ATLAS dataset when a very low number of slices per volume is annotated. When annotation density falls below 20%, prioritizing a higher volume count over annotation density does not result in improved model performance. Our preliminary data on the topic points to the likely impact of the network architecture (the generally observed trend appears to be maintained when AttentionUNet-3D (Oktay et al., 2018) is used in place of UNETR) and a potential impact of an interaction between this dataset and the architecture. More experiments are needed to further elucidate this exception.

We observed that mask interpolation was generally ineffective, except under certain conditions. In most 2D segmentation models, performance remained largely unchanged, with the notable exception of ATLAS at low annotation density. This outcome was anticipated since the pseudo-masks produced via mask interpolation were of relatively low quality (IoU = 0.5621), thereby limiting the upper bound of achievable performance. Intriguingly, in the 3D setting, where baseline performance without mask interpolation was notably suboptimal, the additional spatial context provided by interpolation offset the inherent limitations of the pseudo-masks, resulting in improved outcomes for ATLAS. This suggests that mask interpolation can be advantageous for specific object types when sparse annotations alone fail to yield satisfactory results. Aside from ATLAS, the use of M.I. results in a significant drop in model performance on FGT, possibly due to the loss of tissue details in interpolated masks, such as gaps and spaces in tree-like structures. However, the reason for the absence of a performance drop in 2D cases remains unclear.

Regarding slice selection methods, we did not find any one method to be significantly superior to the others. All three tested methods produced comparable results. Although previous literature suggests that slice selection by UAL-based algorithms can outperform that at fixed interval or random, this advantage was not obvious. We found that UAL tends to pick visually distinct slices, a criteria which may not be the most relevant to the specific segmentation tasks. Additionally, and possibly more importantly, we applied UAL-based selection within each volume to keep the experiment uniform and reduce computational load, rather than across the entire dataset. This practice could have limited UAL's effectiveness compared to dataset-level selection, where the latter might better identify and exclude less informative volumes.

Our findings recommend that in a non-interactive setting, annotation budgets should be unbiasedly distributed across as many volumes as possible for 2D segmentation tasks. While for 3D cases, maintaining a minimum number of annotated slices per volume to ensure effective model training and convergence is also importance. Since all examined slice selection methods yielded comparable results, slice selection at fixed interval is a safe choice, unless the researcher have abundant computing resources to perform iterative active learning or UAL across all slices in the dataset. In our study, we avoided selection methods that are highly biased or radical such as annotating only the initial slices in favor of an even distribution of annotated slices throughout the volume. Future research may examine the impact of deviating from this strategy, for instance, by selecting slices exclusively from a single lesion or even providing only one or two slices per volume, and how such practices affect the convergence of 3D models.

Finally, as mentioned earlier, a more rigorous investigation to precisely estimate annotation costs under different configurations would be beneficial. Nonetheless, our results remain robust even if future studies reveal significant errors in our time estimates. Fortunately, given our experimental setup, any adjustment in annotation cost would simply recalibrate the x-coordinates of our data points without the need to redo the entire analysis.

## Acknowledgments

Research reported in this publication was supported by the National Heart, Lung, and Blood Institute of the National Institutes of Health under Award Number R44HL152825. The content is solely the responsibility of the authors and does not necessarily represent the official views of the National Institutes of Health.

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

## Appendix A. Visualization of selected targets

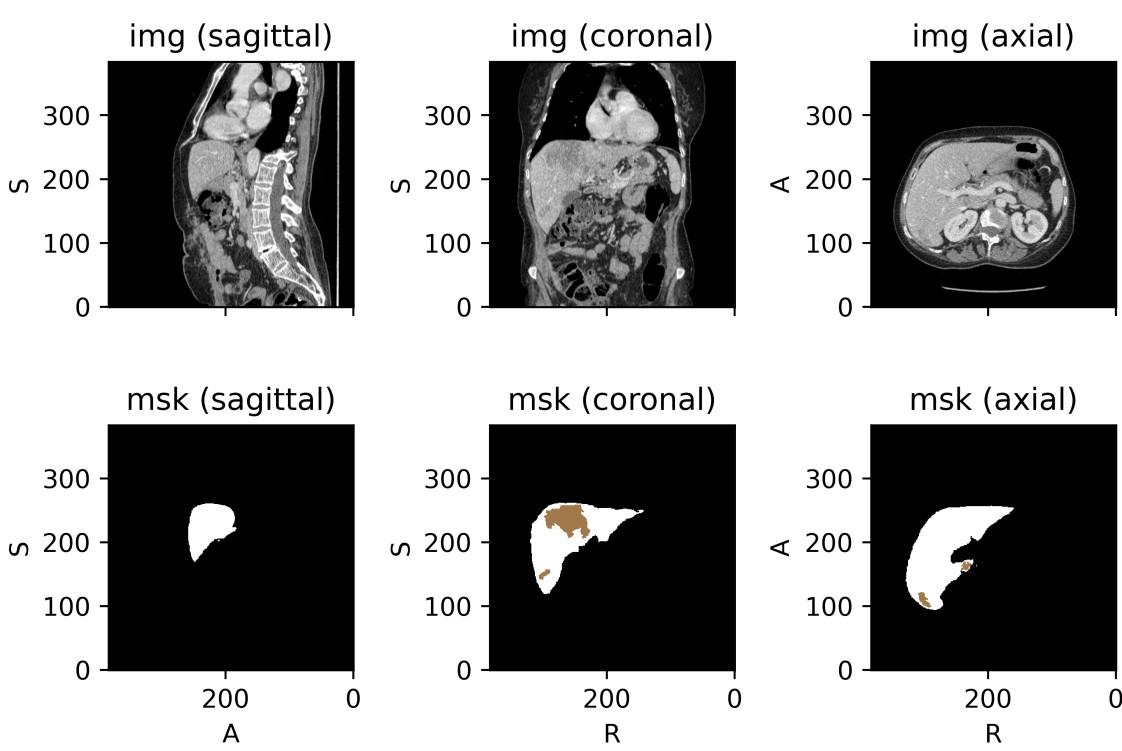

Figure 6: LiTS17: The white masks indicate the liver, while the brown masks indicate liver tumors.

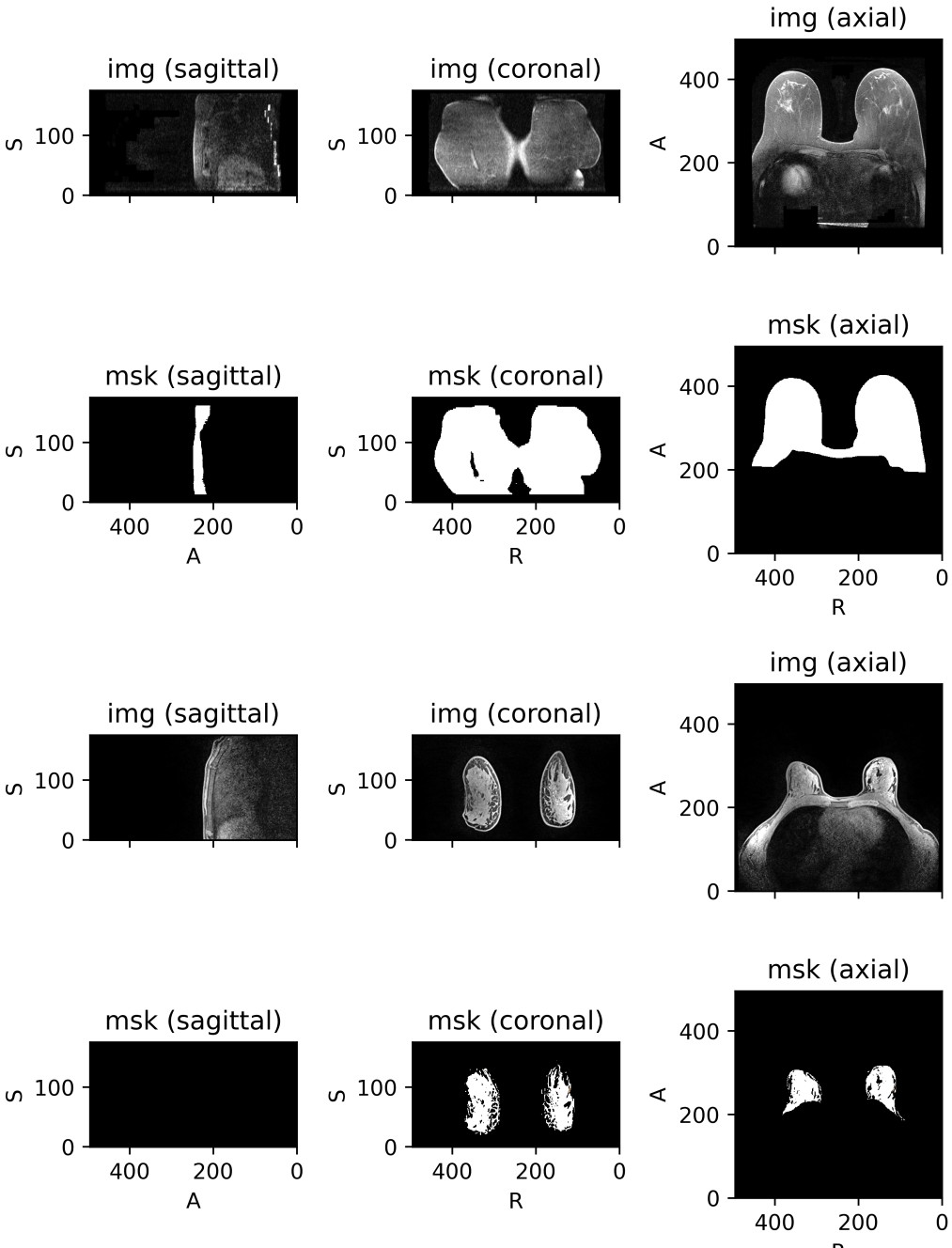

Figure 7: DBC: The top two rows display an MRI scan with corresponding breast masks, while the bottom two rows illustrate an example with fibroglandular tissue masks.

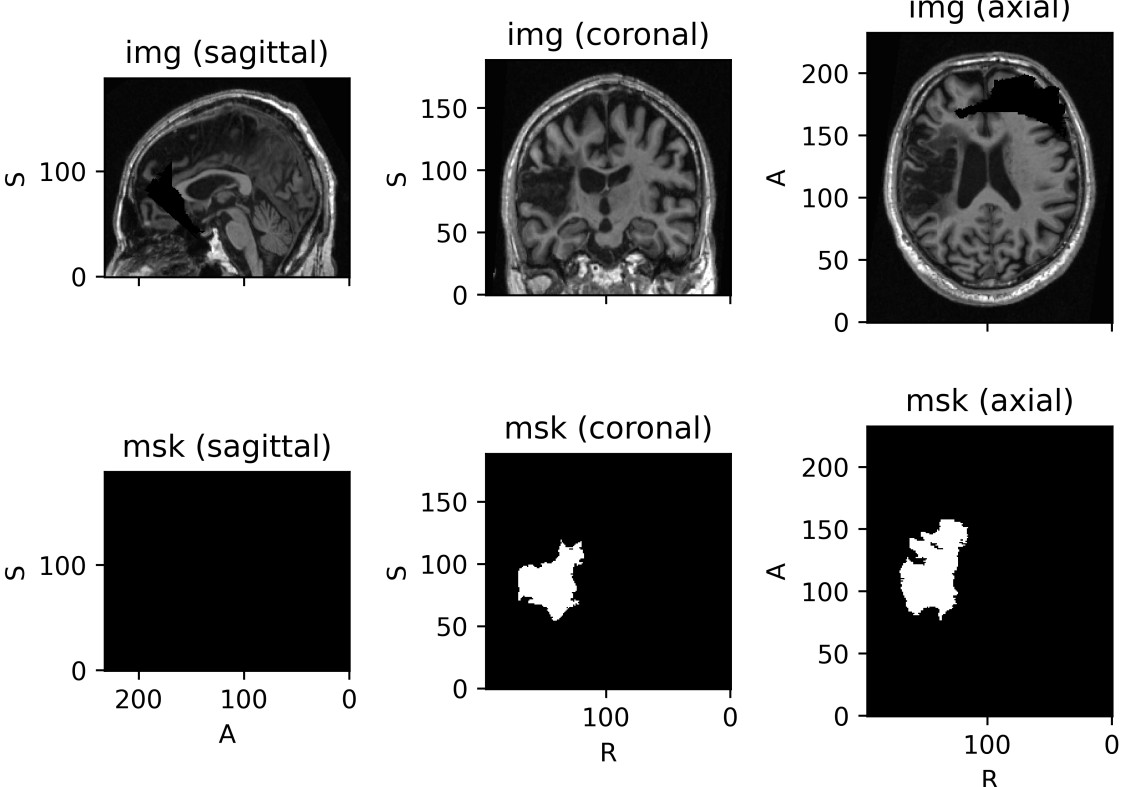

Figure 8: ATLAS: Brain MRI and mask for stroke.

## Appendix B. Rationale of Dataset Selection

The datasets used in this study were deliberately chosen to cover a diverse range of medical imaging characteristics to ensure the generalizability of our findings. The datasets were chosen due to their distinct properties:

- LiTS17: This dataset includes two annotated targets: liver and liver tumor. Our model is trained to segment only the liver tumor class. This dataset features lesion-like objects with strong textural contrast against the background and a narrow range of Hounsfield Unit (HU) intensities.

- DBC: This dataset has three annotated targets (breast mask, vessel, and fibroglandular tissue). We evaluate our model on two tasks: breast mask segmentation and fibroglandular tissue segmentation.

  1. Breast: Organ-like structure defined by clear and smooth edges.
  2. Fibroglandular tissue: Tree-like structure that may or may not have strong contrast against surrounding tissue, adding complexity.

- ATLAS: A large dataset of Brain MRI scans for Anatomical Trace of Lesion After Stroke, making it more suitable for observing performance saturation. The target re-

gions have subtle voxel intensity differences at a local scale, requiring global contextual information for accurate segmentation.

These datasets span various modalities, object types, and levels of structural complexity, allowing us to validate our findings across a broad range of medical imaging scenarios. While no selection can capture all possible cases, we believe our conclusions would generalize to other datasets that present similar segmentation challenges.

## Appendix C. Data Used in the Main Figures with the Statistics

In this appendix, we compile all numerical values used in our primary experiments. Please see Table 2 and 3 for details.

Table 2: Performance (IoU) of 2D models with each annotation configuration

| Volume Count | Density | LiTS17 | DBC | FGT | ATLAS |
|---|---|---|---|---|---|
| 1 | 1 | 0.4957 ± 0.0055 | 0.9345 ± 0.0010 | 0.7114 ± 0.0018 | 0.6454 ± 0.0005 |
| 1 | 0.05 | 0.4867 ± 0.0091 | 0.9269 ± 0.0021 | 0.6974 ± 0.0022 | 0.6301 ± 0.0088 |
| 1 | 0.1 | 0.4937 ± 0.0080 | 0.9293 ± 0.0018 | 0.7060 ± 0.0038 | 0.6515 ± 0.0074 |
| 1 | 0.2 | 0.4990 ± 0.0068 | 0.9268 ± 0.0052 | 0.7098 ± 0.0022 | 0.6524 ± 0.0075 |
| 1 | 0.4 | 0.5005 ± 0.0061 | 0.9286 ± 0.0035 | 0.7116 ± 0.0015 | 0.6428 ± 0.0125 |
| 1 | 0.8 | 0.4924 ± 0.0059 | 0.9307 ± 0.0028 | 0.7136 ± 0.0011 | 0.6467 ± 0.0075 |
| 0.5 | 0.05 | 0.4487 ± 0.0116 | 0.9222 ± 0.0027 | 0.6689 ± 0.0041 | 0.6065 ± 0.0097 |
| 0.5 | 0.1 | 0.4690 ± 0.0092 | 0.9239 ± 0.0015 | 0.6728 ± 0.0066 | 0.6329 ± 0.0080 |
| 0.5 | 0.2 | 0.4757 ± 0.0070 | 0.9189 ± 0.0076 | 0.6788 ± 0.0038 | 0.6431 ± 0.0082 |
| 0.5 | 0.4 | 0.4787 ± 0.0054 | 0.9216 ± 0.0049 | 0.6813 ± 0.0034 | 0.6428 ± 0.0071 |
| 0.5 | 0.8 | 0.4778 ± 0.0074 | 0.9239 ± 0.0035 | 0.6861 ± 0.0072 | 0.6422 ± 0.0067 |
| 0.25 | 0.05 | 0.3517 ± 0.0151 | 0.9028 ± 0.0102 | 0.6423 ± 0.0108 | 0.5500 ± 0.0133 |
| 0.25 | 0.1 | 0.3946 ± 0.0129 | 0.9107 ± 0.0043 | 0.6538 ± 0.0076 | 0.5963 ± 0.0108 |
| 0.25 | 0.2 | 0.3985 ± 0.0184 | 0.9132 ± 0.0040 | 0.6595 ± 0.0100 | 0.6010 ± 0.0138 |
| 0.25 | 0.4 | 0.4043 ± 0.0134 | 0.9142 ± 0.0045 | 0.6652 ± 0.0058 | 0.6123 ± 0.0131 |
| 0.25 | 0.8 | 0.4065 ± 0.0109 | 0.9173 ± 0.0045 | 0.6685 ± 0.0058 | 0.6063 ± 0.0106 |
| 0.125 | 0.05 | 0.2998 ± 0.0296 | 0.9032 ± 0.0180 | 0.6031 ± 0.0112 | 0.5072 ± 0.0271 |
| 0.125 | 0.1 | 0.3202 ± 0.0173 | 0.9106 ± 0.0162 | 0.6320 ± 0.0080 | 0.5504 ± 0.0122 |
| 0.125 | 0.2 | 0.3408 ± 0.0171 | 0.9136 ± 0.0157 | 0.6332 ± 0.0116 | 0.5697 ± 0.0210 |
| 0.125 | 0.4 | 0.3094 ± 0.0240 | 0.9104 ± 0.0078 | 0.6396 ± 0.0093 | 0.5755 ± 0.0203 |
| 0.125 | 0.8 | 0.3024 ± 0.0216 | 0.9153 ± 0.0041 | 0.6363 ± 0.0077 | 0.5656 ± 0.0172 |

Table 3: Performance (IoU) of 3D models with each annotation configuration

| Volume cnt | Density | LiTS17 | DBC | FGT | ATLAS |
|---|---|---|---|---|---|
| 1 | 1 | 0.5710 ± 0.0301 | 0.9143 ± 0.0015 | 0.7195 ± 0.0054 | 0.5817 ± 0.0021 |
| 1 | 0.05 | 0.5363 ± 0.0242 | 0.9089 ± 0.0041 | 0.7107 ± 0.0038 | N/A ± N/A |
| 1 | 0.1 | 0.5573 ± 0.0474 | 0.9100 ± 0.0036 | 0.7127 ± 0.0054 | 0.2133 ± 0.0505 |
| 1 | 0.2 | 0.5689 ± 0.0430 | 0.9148 ± 0.0030 | 0.7161 ± 0.0042 | 0.4663 ± 0.0205 |
| 1 | 0.4 | 0.5643 ± 0.0401 | 0.9122 ± 0.0044 | 0.7130 ± 0.0059 | 0.5360 ± 0.0531 |
| 1 | 0.8 | 0.5776 ± 0.0463 | 0.9150 ± 0.0024 | 0.7166 ± 0.0051 | 0.5758 ± 0.0266 |
| 0.5 | 0.05 | 0.4964 ± 0.0276 | 0.8994 ± 0.0052 | 0.6801 ± 0.0057 | N/A ± N/A |
| 0.5 | 0.1 | 0.4924 ± 0.0232 | 0.9075 ± 0.0026 | 0.6873 ± 0.0080 | 0.1881 ± 0.0380 |
| 0.5 | 0.2 | 0.5099 ± 0.0305 | 0.9094 ± 0.0025 | 0.6927 ± 0.0040 | 0.4195 ± 0.0311 |
| 0.5 | 0.4 | 0.5237 ± 0.0186 | 0.9056 ± 0.0036 | 0.6915 ± 0.0076 | 0.4794 ± 0.0555 |
| 0.5 | 0.8 | 0.5281 ± 0.0358 | 0.9090 ± 0.0022 | 0.6974 ± 0.0051 | 0.5393 ± 0.0145 |
| 0.25 | 0.05 | 0.3716 ± 0.0349 | 0.8861 ± 0.0102 | 0.6621 ± 0.0113 | N/A ± N/A |
| 0.25 | 0.1 | 0.3607 ± 0.0336 | 0.8935 ± 0.0037 | 0.6681 ± 0.0071 | 0.1220 ± 0.0336 |
| 0.25 | 0.2 | 0.4005 ± 0.0408 | 0.8911 ± 0.0041 | 0.6738 ± 0.0076 | 0.3374 ± 0.0231 |
| 0.25 | 0.4 | 0.3957 ± 0.0361 | 0.8911 ± 0.0057 | 0.6783 ± 0.0095 | 0.4023 ± 0.0427 |
| 0.25 | 0.8 | 0.4248 ± 0.0318 | 0.8912 ± 0.0033 | 0.6755 ± 0.0095 | 0.4330 ± 0.0232 |
| 0.125 | 0.05 | 0.2516 ± 0.0651 | 0.8435 ± 0.0101 | 0.6001 ± 0.0123 | N/A ± N/A |
| 0.125 | 0.1 | 0.2606 ± 0.0502 | 0.8565 ± 0.0122 | 0.6142 ± 0.0122 | 0.0309 ± 0.0627 |
| 0.125 | 0.2 | 0.2960 ± 0.0216 | 0.8450 ± 0.0198 | 0.6297 ± 0.0122 | 0.2246 ± 0.0458 |
| 0.125 | 0.4 | 0.2795 ± 0.0321 | 0.8462 ± 0.0143 | 0.6303 ± 0.0170 | 0.2870 ± 0.0474 |
| 0.125 | 0.8 | 0.2722 ± 0.0254 | 0.8505 ± 0.0061 | 0.6239 ± 0.0151 | 0.3239 ± 0.0400 |

## Appendix D.  Numerical results of Model Performance when Trained with Mask Interpolation

| | LiTS17 2D | | | | | LiTS17 3D | | | | |
|---|---|---|---|---|---|---|---|---|---|---|
| $\rho$ | 0.05 | 0.1 | 0.2 | 0.4 | 0.8 | 0.05 | 0.1 | 0.2 | 0.4 | 0.8 |
| $\mu_{sparse}$ | 0.4867 | 0.4937 | 0.4990 | 0.5005 | 0.4924 | 0.5363 | 0.5573 | 0.5689 | 0.5643 | 0.5776 |
| $\sigma_{sparse}$ | 0.0091 | 0.0080 | 0.0068 | 0.0061 | 0.0059 | 0.0242 | 0.0474 | 0.0430 | 0.0401 | 0.0463 |
| $\mu_{MI}$ | 0.4760 | 0.4982 | 0.5023 | 0.4875 | 0.4927 | 0.4984 | 0.5070 | 0.5182 | 0.5076 | 0.5173 |
| $\sigma_{MI}$ | 0.0029 | 0.0021 | 0.0015 | 0.0068 | 0.0058 | 0.0069 | 0.0031 | 0.0089 | 0.0177 | 0.0075 |
| | DBC 2D | | | | | DBC 3D | | | | |
| $\rho$ | 0.05 | 0.1 | 0.2 | 0.4 | 0.8 | 0.05 | 0.1 | 0.2 | 0.4 | 0.8 |
| $\mu_{sparse}$ | 0.9262 | 0.9288 | 0.9306 | 0.9314 | 0.9316 | 0.9089 | 0.9100 | 0.9148 | 0.9122 | 0.9150 |
| $\sigma_{sparse}$ | 0.0016 | 0.0014 | 0.0027 | 0.0028 | 0.0017 | 0.0041 | 0.0036 | 0.0030 | 0.0044 | 0.0024 |
| $\mu_{MI}$ | 0.9264 | 0.9297 | 0.9307 | 0.9322 | 0.9328 | 0.9106 | 0.9132 | 0.9156 | 0.9178 | 0.9141 |
| $\sigma_{MI}$ | 0.0029 | 0.0009 | 0.0017 | 0.0011 | 0.0008 | 0.0025 | 0.0009 | 0.0018 | 0.0024 | 0.0008 |
| | FGT 2D | | | | | FGT 3D | | | | |
| $\rho$ | 0.05 | 0.1 | 0.2 | 0.4 | 0.8 | 0.05 | 0.1 | 0.2 | 0.4 | 0.8 |
| $\mu_{sparse}$ | 0.6974 | 0.7060 | 0.7098 | 0.7116 | 0.7136 | 0.7107 | 0.7127 | 0.7161 | 0.7130 | 0.7166 |
| $\sigma_{sparse}$ | 0.0022 | 0.0038 | 0.0022 | 0.0015 | 0.0011 | 0.0038 | 0.0054 | 0.0042 | 0.0059 | 0.0051 |
| $\mu_{MI}$ | 0.6862 | 0.7100 | 0.7136 | 0.7144 | 0.7125 | 0.5350 | 0.5144 | 0.5301 | 0.5476 | 0.5572 |
| $\sigma_{MI}$ | 0.0018 | 0.0027 | 0.0012 | 0.0021 | 0.0009 | 0.0234 | 0.0051 | 0.0068 | 0.0034 | 0.0173 |
| | ATLAS 2D | | | | | ATLAS 3D | | | | |
| $\rho$ | 0.05 | 0.1 | 0.2 | 0.4 | 0.8 | 0.05 | 0.1 | 0.2 | 0.4 | 0.8 |
| $\mu_{sparse}$ | 0.6313 | 0.6419 | 0.6375 | 0.6395 | N/A | 0.4487 | 0.5189 | 0.5692 | 0.5857 | N/A |
| $\sigma_{sparse}$ | 0.0047 | 0.0059 | 0.0072 | 0.0091 | N/A | 0.0294 | 0.0059 | 0.0088 | 0.0053 | N/A |
| $\mu_{MI}$ | 0.5463 | 0.6225 | 0.6350 | 0.6518 | N/A | 0.5019 | 0.5524 | 0.5774 | 0.5621 | N/A |
| $\sigma_{MI}$ | 0.0031 | 0.0092 | 0.0049 | 0.0020 | N/A | 0.0192 | 0.0121 | 0.0146 | 0.0051 | N/A |

Table 4: This table presents the numerical results (mean and standard deviation) of model performance when trained with and without the use of M.I. across different annotation densities ($\rho$). For the ATLAS dataset, weighted sampling was employed to select slices for annotation, ensuring that the interpolated masks maintained sufficient quality for a meaningful performance comparison. The configuration with $\rho = 0.8$ is omitted, as at this density, all positive slices would have already been pre-selected, leaving M.I. non-applicable.

## Appendix E.  Quality of Interpolated Masks

After mask interpolation, the resultant annotation is a mixture of human-created annotation and algorithm-generated mask for slices without human annotation. Their quality is measured as the batched IoU between the resultant annotated derived from human-generated

sparse annotation and the densely annotated counterpart. We report their quality across different annotation densities.

| Annotation Density ($\rho$) | 0.05 | 0.1 | 0.2 | 0.4 | 0.8 |
|---|---|---|---|---|---|
| **LiTS17 (Tumor)** | 0.9079 | 0.9431 | 0.9683 | 0.9931 | 0.9926 |
| **DBC (Breast)** | 0.9673 | 0.9781 | 0.9856 | 0.9912 | 0.9971 |
| **DBC (FGT)** | 0.6366 | 0.7118 | 0.7763 | 0.8447 | 0.9456 |
| **ATLAS (Trace of Stroke)** | 0.5621 | 0.7326 | 0.8831 | 0.9674 | N/A |

Table 5: Quality of dense annotation interpolated from sparse annotation at different densities ($\rho$). Please check the caption of Table 4 for the "N/A" in this table.

## Appendix F. Algorithm for Generating Dense Annotation from Sparse Annotation with Mask Interpolation

### F.1. Implementation and Training of M.I.

Our implementation of the Mask Interpolation algorithm incorporates elements from both Sli2Vol (Yeung et al., 2021) and SSA (Wu et al., 2022). These recent works were selected because they reported superior interpolation performance and provided partial code.

Following Sli2Vol's setup, we use a ResNet18 encoder as the feature extractor to derive features from adjacent slices and the original reconstruction module provided in the official GitHub repository. Thus, the model architecture in our Mask Interpolation (M.I.) algorithm is identical to that proposed by Sli2Vol. However, while the original Sli2Vol implementation was trained solely on adjacent slices, Wu et al. demonstrated that interpolation across larger gaps is also feasible. Therefore, for each volume, we randomly sample K slices, sort them by index, and train the model to interpolate from the nearest slices with smaller or larger indices.

Since inference is performed in a cascading manner, error accumulation can degrade performance. Sli2Vol proposed a verification module based on pixel intensity to address this issue, but its implementation was not provided. Additionally, due to its rule-based and intensity-dependent nature, we were concerned about its potential sensitivity to brightness variations. To mitigate error accumulation during inference, we adopted the scheduled sampling strategy from SSA, which replaces some ground truth reference slices with reconstructed slices during training.

Our implementation achieved similar Mask Interpolation (M.I.) performance on LiTS17-Liver as reported in the Sli2Vol and SSA papers. This consistency gives us confidence in the correctness of our implementation.

### F.2. Inference of M.I.

See Algorithm 1.

---

**Algorithm 1** Generating Dense Annotation from Sparse Annotation with M.I.

---

1: $pos\_queue \leftarrow \{\text{human annotated masks}\}$
2: **while** $pos\_queue$ is not empty **do**
3:      $mask_i \leftarrow pos\_queue.dequeue()$
4:      $mask_{i-1}, mask_{i+1} \leftarrow mask\_prop(slices_i, mask_i)$
5:      **if** $slice_{i-1}$ is new and $slice_{i-1}$ contains object **then**
6:          $pos\_queue.addback(mask_{i-1})$
7:      **end if**
8:      **if** $slice_{i+1}$ is new and $slice_{i+1}$ contains object **then**
9:          $pos\_queue.addback(mask_{i+1})$
10:      **end if**
11: **end while**
12: Assign 0 to the rest of the unannotated slices.

---

## Appendix G. Details for Model Training

We selected various variants of representative segmentation models commonly used in the computer vision community and paired them with each dataset based on its level of difficulty. In other words, more challenging datasets were matched with models having larger parameter sizes and greater prediction capabilities. This approach allows us to cover a wide range of training settings and datasets without having to repeat experiments across all possible (dataset-model) combinations.

When training 2D segmentation models, samples in each batch are drawn uniformly at random from the annotated slices. We used a batch size equals to 16 for all datasets.

1. LiTS17 is paired with Inception-UNet (Punn and Agarwal, 2020) for liver tumor segmentation, with Random Bias Field, Random Elastic Deformation, Random Flip as augmentation. We used weighted Cross Entropy Loss with weight [0.04, 0.25, 1.4] for the background, liver and tumor classes. We train the model for 14000 iterations using AdamW optimized with lr = 1e-4, weight-decay = 1e-5, and a polynomial learning rate scheduler with power set to 0.8.

2. DBC is paired with SegFormer-B3 (Xie et al., 2021) initialized on MiT-B3 weight for both the breast segmentation and fibroglandular tissue segmentation tasks. We converted gray scale images to RGB and performed Random Flip as augmentation. We used Cross Entropy loss and train the model with AdamW optimizer for 12000 iterations. The optimizer hyper-parameters were set to lr = 1e-4, weight_decay = 1e-2, with a polynomial learning rate scheduler with power set to 0.8.

3. ATLAS is paired with CAM-WNet (Liu et al., 2022), with VGG16 encoder in the coarse-segmentation module initialized from weights pretrained on ImageNet. Random Flip was used as augmentation. The loss function used a combination of Soft Dice Loss and Cross Entropy Loss. The model is trained with AdamW optimizer with lr = 5e-5, weight_decay = 1e-5. for 15000 iterations. The learning rate is multiplied by 0.7 per 2000 iterations.

When training 3D segmentation models, a patch-based trained framework was adopted. All models were trained from scratch without pre-initialization.

1. LiTS17 is paired with 3D-UNet (Çiçek et al., 2016) for liver tumor segmentation, with Random Bias Field, Random Affine, Random Flip as augmentation. We used weighted Cross Entropy Loss with weight [0.04, 0.2, 1.0] for the background, liver and tumor classes. We train the model for 18000 iterations with each batch consists of 24 patches and each patch has dimension $128 \times 128 \times 128$ then down-sampled to $64 \times 64 \times 64$. Such patches are sampled in a weighted manner so they centers with a 50% probability on liver, 30% on tumor, and 20% on background and unannotated voxels combined to focus on the areas of interest. We used SGD optimized with lr = 0.01, weight-decay = 1e-4, momentum = 0.9 without any learning rate scheduling.

2. DBC is paired with SegResNet (Myronenko, 2019) for both breast segmentation and fibroglandular tissue segmentation. We used Random Affine and Random Flip as augmentation. We used Cross Entropy loss and train the model with AdamW optimizer for 12000 iterations. The batch size was set to 24 with each patch being $128 \times 128 \times 64$ sub-volumes randomly sampled from the full volume. The optimizer hyper-parameters were set to lr = 3e-4, weight_decay = 1e-5, with Cosine Annealing learning rate scheduling.

3. ATLAS is paired with UNETR (Hatamizadeh et al., 2022). We found this combination to benefit from larger batch size, so each batch consists of 54 patches. Each $96 \times 96 \times 64$ patch is sampled with a probability of 30% for background, 55% for lesions, and 15% for unannotated voxels to be in the patch center. Random Affine and Random Flip are used as augmentation. The loss function used a combination of Soft Dice Loss and Focal Loss. The model is trained with AdamW optimizer with lr = 1.5e-4, weight_decay = 1e-6 for 15000 iterations. The learning rate is multiplied by 0.7 per 3000 iterations.

