# OpenReview forum: "How to select slices for annotation to train best-performing deep learning segmentation models for cross-sectional medical images?"
_MIDL.io/2025/Conference — MIDL 2025 Poster_

### Official Review · Reviewer_h1uM · 2025-02-14

**Confidence:** 5
**Preliminary Rating:** 3
**Final Rating:** 4

**Summary:**

In the paper, the authors investigated how to select slices of 3D medical images to maximize performance of the resulting deep learning segmentation models. The variables are annotation budgets, numbers of annotated cases, numbers of annotated
slices per volume, slice selection techniques, and mask interpolations.

**Strengths:**

The authors’ research topic is interesting and targeting to a practical problem. It may be provided as insights when some clinical institutions have limited budget and would like to do AI-related research.

**Weaknesses:**

1. The authors have too strong assumptions in the experimental design, especially the 3 & 4 & 5. 3) that the datasets have consistent physical spacing, which may not hold for majority of the 3D medical images due to physical constraints. 4) the number of slides needed to be annotated are quite different if you want to segment lesions versus organs. 5) the cost needed to be annotated are quite different if you want to segment lesions versus organs. These limits the generalizability of the work a lot. For example, for 3), is it valid if you assume there are same number of slides for prostate cancer lesions annotation and breast FGT annotation? Or the bones of thigh and the liver cancers?
2. The sparse annotation strategy will miss small lesions, and thus the design might not that generalizable002E

**Detailed Comments:**

Please see other sections.

**Justification Of The Final Rating:**

The authors responded my comments well, and I have no more comments to made. I have edited my rating correspondingly. The authors responded my comments well, and I have no more comments to made. I have edited my rating correspondingly.

**Justification Of The Preliminary Rating:**

Interesting and practical topic, but the strong assumptions made by the authors led the conclusions highly not generalizable. I would like to hear back from the authors and also learn from other reviewers' opinions.

**Questions To Address In The Rebuttal:**

1. Table 1, I would like to ask the authors to provide more specific context about the primary targets of each dataset. You have 4 primary targets, but 3 datasets. It is hard to follow.
2. Who are the people who did the annotations? Are they radiologists, or trainees? How to do control on the variance of the annotators?
3. Why different datasets used different DL models? I think this makes the experimentations conducted more non-comparable.
4. Please also response to the opinions listed in weakness.

---

> ### Author Response · Authors · 2025-03-07
> **We appreciate the reviewer's opinions and we provide a point-to-point response to the reviewer's concern**
>
> ## Weakness
>
> **1 (3): the datasets have consistent physical spacing, which may not hold for majority of the 3D medical images due to physical constraints.**
>
> In our study, we apply resampling and crop out slices with excessive irrelevant body part as part of the image pre-processing pipeline to ensure each voxel has consistent spacing and each volume the same dimensions. This approach is nearly lossless, requires minimal or no human intervention and would not interfere with the image collection process.
>
> **1 (4): the number of slides needed to be annotated are quite different if you want to segment lesions versus organs.**
>
> We agree that the number of slides needed to satisfactorily segment different targets can vary significantly. However, our study tested a wide range of annotation budget, from annotating 1/160 of the slices to annotating all available slices within the dataset. Given this wide coverage of different budget levels, we believe our conclusions remain meaningful and valid across various annotation scenarios.
>
> **1 (5): the cost needed to be annotated are quite different if you want to segment lesions versus organs. These limits the generalizability of the work a lot. For example, for 3), is it valid if you assume there are same number of slides for prostate cancer lesions annotation and breast FGT annotation? Or the bones of thigh and the liver cancers?**
>
> We appreciate the reviewer’s concern. Since both Assumption (3) and (5) were mentioned, and we have justified (3), we will focus here on Assumption (5).  We believe annotation cost can be decomposed into two components:
>
> [1] the time needed to identify/localize the target and
>
> [2] the time needed to find the contour and draw or refine the mask.
>
> For component [2], the time required for annotating a batch of slices sampled in a nearly unbiased manner should be similar across batches (as the ratio of slice containing the target converges in probability to the ratio of slices containing target in the dataset). For component [1], at lower annotation densities—where selected slices are unlikely to capture the same object—the localization time should also be roughly proportional to the number of slices annotated. On the other hand, for easily distinguishable organ-like structures such as bones, [1] is expected to be minimal, as experienced annotators can identify the target almost instantly.
>
> Based on this rationale, we believe our assumption remains valid in most scenarios.
>
> **2: The sparse annotation strategy will miss small lesions, and thus the design might not that generalizable.**
>
> We appreciate the reviewer’s concern regarding the potential loss of small lesions due to sparse annotation. Our annotation strategy mitigates this issue by assigning an "ignore" label to unannotated slices rather than treating them as negative class, which is a standard operation in segmentation (used by VOC, COCO) where precise annotation cannot be satisfactorily obtained. This ensures that missing small lesions would not introduce false supervision signals during training.
>
> ## Questions To Address In The Rebuttal
>
> **1: Table 1, I would like to ask the authors to provide more specific context about the primary targets of each dataset. You have 4 primary targets, but 3 datasets. It is hard to follow.**
>
> As per the reviewer’s request for additional clarification regarding the primary targets in each dataset. We add a more detailed description on the rational of choosing these datasets and a visualization of each dataset/target in the appendix. We also changed Table 1 to reflect all datasets/targets involved in our study.
>
> **2: Who are the people who did the annotations? Are they radiologists, or trainees? How to do control on the variance of the annotators?**
>
> Our study is retrospective in nature, and all sparse annotations are derived from publicly available fully annotated datasets. According to the original dataset documentation, all annotations were either performed or reviewed and approved by radiologists for quality control.
>
> **3: Why different datasets used different DL models? I think this makes the experimentations conducted more non-comparable.**
>
> We acknowledge the reviewer’s concern about the use of different deep learning models for different datasets. The purpose of our study is not to identify the best-performing model architecture across all datasets but rather to validate our conclusions across a diverse set of architectures. Our selection of models is constrained by available computational resources, but we ensure that each model’s performance aligns with results reported in mainstream literature. This approach allows us to demonstrate that our findings would hold consistently across different datasets and model choices.

---

### Official Review · Reviewer_VRnB · 2025-02-23

**Confidence:** 4
**Preliminary Rating:** 3
**Recommendation:** Poster
**Final Rating:** 4

**Summary:**

The manuscript explores a crucial question: how to optimize annotation for cross-sectional medical images within a limited budget. It conducts experiments on three datasets across four segmentation tasks and provides an in-depth discussion of the topic.

**Strengths:**

- The topic is meaningful to the development of deep learning models for 3D medical image segmentation.
- The experiments cover a wide range of factors (4 tasks, both 2D and 3D models), and results are well represented.
- The results generally align with the conclusions of the paper.

**Weaknesses:**

- While the paper addresses a practical question in the real-world development of deep learning models for segmentation, its conclusions largely reflect the established practices of the field based on practitioners' experience, limiting its contribution.
- The segmentation performance was evaluated solely using IoU. Incorporating additional metrics, such as Hausdorff distance, would provide a more comprehensive assessment.

**Detailed Comments:**

- What's the motivation of using different model architectures for different datasets?
- What's the motivation of choosing the datasets? How representative are they? Can the conclusions drawn from these datasets be generalize to other data?
- Were there any data augmentations used in model training?

**Justification Of The Final Rating:**

I would like to thank the authors for their feedback. I agree with the authors that the study provides empirical validation of certain de facto practices, which is crucial for transitioning from anecdotal best practices to rigorously tested principles. I also appreciate the authors' efforts in benchmarking across diverse datasets. Therefore, I have raised my rating to 4.

**Justification Of The Preliminary Rating:**

The paper studies an important question with relatively comprehensive experiments and well-represented results. However, I think its conclusions largely reflect the established practices of the field based on practitioners' experience, limiting its contribution.

**Questions To Address In The Rebuttal:**

Please see Weaknesses and Detailed Comments.

---

> ### Author Response · Authors · 2025-03-07
> **We thank the reviewer for their thoughtful feedback and suggestions. Here is a point-to-point response to the reviewer's comments**
>
> **While the paper addresses a practical question in the real-world development of deep learning models for segmentation, its conclusions largely reflect the established practices of the field based on practitioners' experience, limiting its contribution.**
>
> We appreciate the reviewer’s feedback and agree that some of our conclusions may align with established practices. However, our study provides empirical validation of these practices, which is critical in moving from anecdotal best practices to rigorously tested principles. While many segmentation practitioners may have intuitive expectations regarding annotation trade-offs, there has been a lack of systematic, quantitative studies to confirm or challenge these assumptions across different datasets and model architectures.
>
> **The segmentation performance was evaluated solely using IoU. Incorporating additional metrics, such as Hausdorff distance, would provide a more comprehensive assessment.**
>
> We agree with the reviewer that including the HD95 metric would be beneficial. However, due to the large volume of experiments conducted in this study, it is infeasible for us to redo the whole experiments to compute HD95 at this time. That said, our primary evaluation metric, IoU, is widely accepted for semantic segmentation and can be deterministically converted to Dice score using the formula: $Dice = \dfrac{2 \times IoU}{ IoU + 1}$. Furthermore, prevalent literature has demonstrated a strong correlation between Dice score and HD95, supporting the validity of our evaluation approach. Nonetheless, we acknowledge the importance of HD95 and will explicitly note its absence as a limitation of our work.
>
> **What's the motivation of using different model architectures for different datasets?**
>
> Our choice of models and training setups are intentional to ensure that each model’s performance aligns with results reported in the mainstream literature had they be trained on densely annotated datasets. It also allows us to validate that our findings are not specific to a single model architecture but rather hold consistently across different datasets and model choices.
>
> **What's the motivation of choosing the datasets? How representative are they? Can the conclusions drawn from these datasets be generalize to other data?**
>
> We appreciate the reviewer’s question regarding our dataset selection and its representativeness. Our motivation was to cover a diverse range of medical imaging characteristics to ensure the generalizability of our findings. The datasets were chosen based on their distinct properties and given they span a broad range of image characteristics. Hence, we believe our conclusions would generalize to other data. A more detailed justification of our choice is added to the Appendix under " Rationale of Dataset Selection".
>
> **Were there any data augmentations used in model training?**
>
> Yes, we applied data augmentations during model training. The specific augmentation techniques used are detailed in the Appendix under "Details for Model Training".

---

### Official Review · Reviewer_FNCD · 2025-02-23

**Confidence:** 5
**Preliminary Rating:** 1

**Summary:**

This paper addresses the problem of how to select slices of cross-sectional medical images to optimize deep-learning segmentation models. The authors conduct experiments across three datasets, evaluating different slice selection strategies, annotation densities, and the impact of mask interpolation (M.I.). The study concludes that annotating more volumes with fewer slices per volume is generally preferable, mask interpolation is often ineffective with exception, and unsupervised active learning (UAL) does not provide significant benefits over simpler selection methods such as random selection or selection at fixed interval.

**Strengths:**

- importance of the topic. Image annotation is an essential aspect of training AI models, as the accuracy of the models relies on available labels, and annotation costs constitute a significant bottleneck in training deep learning models
- The paper is well-organized, with figures illustrating key trends in model performance.

**Weaknesses:**

- the core idea of the paper (influence of annotation density, mask interpolation and slice selection) has already been presented in the literature
- The choice of the slice selection strategies and mask interpolation is not explained. The paper compares fixed-interval, random, and UAL-based slice selection but does not justify why these particular methods were chosen.  The mask interpolation technique is evaluated, but the rationale behind the specific implementation and comparison to alternative interpolation methods is unclear.

- Weak Experimental Robustness. Only one train/validation/test split is used for each dataset. This limits the generalizability of the findings, as performance may vary significantly across different splits. No statistical significance testing is reported.

- Three datasets alone are insufficient to claim generalizability across medical imaging applications.

**Detailed Comments:**

- Authors should also consider that annotation costs are not equal across slices and that annotating slices with pathology is typically more time-consuming than annotating normal slices.
- a table with numerical results (average and std)  should be added

**Justification Of The Preliminary Rating:**

The paper lacks novelty, and the study's design (i.e. method, slice interpolation and slice choice) is unclear. Moreover, the experimental part has insufficient robustness, only three datasets are used and no statistical test is provided

**Questions To Address In The Rebuttal:**

- Add cross-validation
- Explain the rationale for the choice of methods,  mask interpolation and slice selection
- Add statistical test, a table with the results (average and standard deviation to the results)

---

> ### Author Response · Authors · 2025-03-07
> **We appreciate the reviewer's suggestions and opinions and provide a point-to-point response to the reviewer's concern**
>
> **the core idea of the paper (influence of annotation density, mask interpolation and slice selection) has already been presented in the literature**
>
> We appreciate the reviewer's opinion. Our study is not intended to introduce algorithms, but to validate common slice selection strategies and annotation trade-offs, which we believe is valuable for the medical AI community. While previous work may exam some components individually or within specific dataset, our approach offers a comprehensive validation analysis. To the best of our knowledge, no prior work has done similar study.
>
> **The choice of the slice selection strategies and mask interpolation is not explained....[Explain the rationale for the choice of methods, mask interpolation and slice selection]**
>
> We appreciate the reviewer’s comment and acknowledge that the detailed rationales were omitted from the main text. For the slice selection strategies, we chose fixed-interval and random methods as rule-based baselines and complemented these with a UAL-based policy. The UAL-based approach respects assumption (2)—that supervised training begins only after the complete set of annotations is available—which reflects a realistic constraint in many annotation pipelines.
>
> Regarding the mask interpolation technique, our implementation combines elements from both Sli2Vol and SSA. These recent works were selected because they reported superior interpolation performance and provided partial code, allowing us to confirm the reproducibility of similar results on the LiTS17 dataset. We have added the associated information in Appendix F.1.
>
> **Only one train/validation/test split ... different splits. No statistical significance testing is reported. [Add cross-validation], [Add statistical test, a table with the results (average and standard deviation to the results)]**
>
> We appreciate the reviewer’s concerns regarding the robustness of our results. However, our chosen train/val/test splitting strategy follows standard practices in the literature and aligns with established benchmarking protocols in computer vision. Additionally, we maintain a consistent test set across all experimental trials for each dataset, ensuring a fair and meaningful comparison across different hyperparameter settings while minimizing variability introduced by data partitioning changes.
>
> That said, we agree with the reviewer's insights regarding statistical results. In response to the reviewer's request, we have included three tables in the appendix, reporting the mean and standard deviation for all experiments in our manuscript. Furthermore, we have added error bars to most applicable figures to better illustrate performance variability.
>
> **Three datasets alone are insufficient to claim generalizability across medical imaging applications.**
>
> We appreciate the reviewer’s concern regarding the generalizability of our findings. While we acknowledge that using three datasets does not capture the full spectrum of medical imaging applications (which is a limitation of our work), the consistency of our findings across datasets of very different nature and model choices provides strong evidence that the observed trends are meaningful and applicable beyond the specific datasets used. We add clarification and justification on the coverage of our dataset choice in Appendix B.
>
> **Authors should also consider that annotation costs are not equal across slices and that annotating slices with pathology is typically more time-consuming than annotating normal slices.**
>
> We agree with the reviewer that annotation costs can vary, and we did not assume equal cost across all slices. An assumption of near-equal cost for a batch of nearly unbiasedly sampled slices (with a mix of slices with and without target objects) is sufficient. We decompose annotation cost into two components: (1) the time needed to identify the target and (2) the time needed to draw or refine the mask.  For component (2), the time required for annotating a batch of slices sampled in a nearly unbiased manner should be similar across batches (as the ratio of slice containing the target converges in probability to the ratio of slices containing target in the dataset). For component (1), at lower annotation densities—where selected slices are unlikely to capture the same object—the localization time should also be roughly proportional to the number of slices annotated. On the other hand, for easily distinguishable organ-like structures such as bones, (1) is expected to be minimal, as experienced annotators can identify the target almost instantly.
>
> Based on this rationale, we believe our assumption holds realistically in most scenarios.

---

> ### Author Response · Authors · 2025-03-14
>
> Dear Reviewer,
>
> Thank you again for your time and feedback on our manuscript. We have addressed the requested revisions. However, we have not heard back regarding the final assessment, and as the discussion period deadline is approaching, we wanted to kindly check if you have any further comments or concerns.

---

### Official Review · Reviewer_oZXG · 2025-02-25

**Confidence:** 4
**Preliminary Rating:** 4
**Recommendation:** Poster
**Final Rating:** 4

**Summary:**

In this study, the authors explore various strategies to establish best practices for selecting an optimal set of slices to annotate for training a DL segmentation model within an annotation budget. Across four datasets, their results show that annotating more volumes with fewer slices leads to higher performance compared to annotating fewer volumes with more slices. The authors also show that naively selecting slices (e.g., randomly) is equivalent to using unsupervised active learning techniques. Finally, they demonstrate that mask interpolation between slices improves performance only in certain cases.

**Strengths:**

- The paper tackles the important challenge of medical image annotation, especially within the constraints of a fixed annotation budget.
- The paper provides valuable insight into various slice selection strategies that will benefit other researchers in the community.
- The experiments across both 2D and 3D settings are systematic and thorough.

**Weaknesses:**

- The paper focuses on exploring slice selection strategies at the volume level rather than the dataset level, which is more relevant for training segmentation models. Existing active learning approaches dynamically select slices within each volume based on its complexity.
- The authors claim that unsupervised active learning techniques are equivalent to random sampling, etc. It is unclear which specific method they are using. Furthermore, detailed comparison with various active learning approaches is warranted.

**Detailed Comments:**

- The discussion of results in Section 3.2 page 7 regarding the potential cause of performance drop when using mask interpolation should be moved to the discussion section.
- It is not clear if the experiments use annotations provided in each dataset or the volumes were reannotated for the purpose of the study. I assume its the former.

**Justification Of The Final Rating:**

The authors have addressed all of my questions. While I acknowledge the concerns raised by other reviewers regarding the assumptions made in this work, I believe the findings provide valuable insights for the MIDL community. Therefore, I recommend acceptance.

**Justification Of The Preliminary Rating:**

The paper provides valuable insights into optimizing slice selection for annotating medical images across multiple datasets. While a more detailed analysis would strengthen the overall conclusions (especially regarding active learning), I believe the findings would benefit other researchers in the MIDL community.

**Questions To Address In The Rebuttal:**

Please see weaknesses above.

---

> ### Author Response · Authors · 2025-03-07
> **We appreciate the reviewer’s valuable insights. We are pleased that the reviewer recognizes the importance of our study and the systematic nature of our experiments. Here, we provide a point-to-point response to the reviewer's concern**
>
> **The paper focuses on exploring slice selection strategies at the volume level rather than the dataset level, which is more relevant for training segmentation models. Existing active learning approaches dynamically select slices within each volume based on its complexity.**
>
> We agree that applying active learning (AL) approaches at the dataset level could, in principle, achieve better performance than random selection. To prevent misinterpretation, we explicitly highlighted in the Discussion section that our finding—"UAL is no better than random selection"—applies only when UAL is performed at the slice level, not the dataset level. This is a limitation of our work. However, applying UAL-based selection at the dataset level is computationally expensive, and traditional (interactive) AL-based selection is even more costly, as both require scanning all unannotated slices before selecting the next sample for annotation. Given these constraints, we explored the impact of UALs on slice selection in a more restricted manner. Despite these limitations, our study provides valuable insights into what does not work, and we acknowledge that a systematic validation of AL at the dataset level would be an important direction for future research.
>
> **The authors claim that unsupervised active learning techniques are equivalent to random sampling, etc. It is unclear which specific method they are using. Furthermore, detailed comparison with various active learning approaches is warranted.**
>
> We used a simplified version of Representative Annotation [1]. Specifically, we first trained a Masked Autoencoder (MAE) for feature extraction and then used these features to compute the representativeness score. This method also aligned with the coreset optimization approach proposed by Sener et al. [2]. We chose Representative Annotation [1] because it laid the foundation for many subsequent unsupervised active learning (UAL) methods. We added this information to section 2.3
>
> While we acknowledge that a detailed comparison with various AL approaches could be insightful, conducting additional experiments in this direction is beyond the primary focus of our study. Our work primarily aims to investigate the trade-off between annotation density and volume count and easily-deployable pre-processing, rather than comprehensively benchmark AL methods. However, we appreciate the reviewer’s perspective and would be happy to enrich our related work and discussion with relevant literature if the reviewer has specific suggestions.
>
> **Reference:**
>
> [1] Zheng, Hao, et al. "Biomedical image segmentation via representative annotation." Proceedings of the AAAI Conference on Artificial Intelligence. Vol. 33. No. 01. 2019.
>
> [2] Sener, Ozan and Silvio Savarese. “A Geometric Approach to Active Learning for Convolutional Neural Networks.” ArXiv abs/1708.00489 (2017): n. pag.
>
> **The discussion of results in Section 3.2 page 7 regarding the potential cause of performance drop when using mask interpolation should be moved to the discussion section.**
>
> In response of the reviewer's request, we have moved the associated texts to the Discussion and Conclusion.
>
> **It is not clear if the experiments use annotations provided in each dataset or the volumes were reannotated for the purpose of the study. I assume its the former.**
>
> Yes, we used the annotations provided in each dataset, as the reviewer correctly assumed.

---

### Official Review · Reviewer_xr3K · 2025-02-26

**Confidence:** 3
**Preliminary Rating:** 4
**Recommendation:** Poster
**Final Rating:** 4

**Summary:**

The authors conduct experiments to determine which slices should be annotated for efficient model training considering number of slices, number of volumes, slice selection strategies and mask interpolation to create dense annotations. The results show that annotated slices should be distributed across as many volumes as possible, and slice selection at a fixed interval is sufficient.

**Strengths:**

* Very nicely structured manuscript, clear explanations.
* Various settings are tested (number of slices, number of volumes, 3 slicing techniques, mask interpolation)
* The authors selected representative segmentation models for each dataset and performed hyper-parameter tuning to ensure a performance comparable to existing literature.

**Weaknesses:**

* The assumption that the cost of annotating each batch of N slices is approx consistent seems to limit realistic scenarios if cost is related to time (which depends on various factors like object size, complexity, ...)
* Although chosing representative segmentation models per dataset is a strength of the manuscript, even the authors state that there might be an interation between dataset, architecture and observations. Without some additional experiments (ablation studies), the question is whether the same observations hold for different architectures (as indicated in Section 4).

**Detailed Comments:**

* Assumption (5) says that the cost for annotating each batch of N slices is approximately consistent. Could the authors clarify what they mean by "cost of annotating"? And if time is considered a cost factor, this would mean that the time for annotating different slices in a volume is somewhat constant (which probably depends on factors like object size, complexity, ...) Could the authors comment on whether this is a realistic assumption for the chosen datasets?
* There is no table 4 (Sectoin 3.2), authors probably meant figure 4?
* Figure 4 ATLAS with M.I. stops at density = 0.4,  are there no values for 0.8?
* For the M.I. algorithms experiments (Figure 4), it would be interesting to see the results for density 1.0 (dense annotations) as comparison (upper limit).
* Appendix C is not mentioned in the main manuscript
* As various datasets are used and some data characteristics are discussed (eg in Section 3.2 FGT), it would help the reader to see some examples of the reference data and the predictions (in the appendix).
* minor comments:
 * Section 4 could be renamed Discussion and Conclusion, as both are combined into one section
* some typos, missing spaces in the appendix

**Justification Of The Final Rating:**

As stated also by other reviewers, the strict assumptions decrease the generalizability. However, as already stated in the preliminary review, I still believe that it is enough for a full paper contribution, as I believe the findings would benefit other researchers in the MIDL community as well.

**Justification Of The Preliminary Rating:**

The experiments performed by the authors give insight into how to chose slices for annotation under some general assumptions, which supports decision-making for annotation generation. The authors should provide some clarification to the raised concerns about the posed assumptions and the need for futher research on the interaction between datasets, architectures and observations. Even with these two points remaining open, I consider the contribution still enough for a full paper contribution, as multiple experiments on different datasets and with various settings have been performed by the authors and very clear observations are concluded.

**Questions To Address In The Rebuttal:**

* Clarifications for assumption (5) (see detailed comments)
* Representative qualitative examples of some experiments
* Why is ATLAs with M.I. for density 0.8 not available in Figure 4?
* minor comments from detailed comments

---

> ### Author Response · Authors · 2025-03-07
> **We appreciate the reviewer’s positive feedback and recognition of our work. We are glad that the manuscript’s structure and our experimental design was found to be appropriate. Here is a point-to-point response to the reviewer's concern.**
>
> **Assumption (5) says that the cost for annotating each batch of N slices is approximately consistent. Could the authors clarify what they mean by "cost of annotating"? And if time is considered a cost factor, this would mean that the time for annotating different slices in a volume is somewhat constant (which probably depends on factors like object size, complexity, ...) Could the authors comment on whether this is a realistic assumption for the chosen datasets? [Clarifications for assumption (5) (see detailed comments)]**
>
> We acknowledge that annotation time can vary based on factors such as object size and complexity, but we don't think it contradicts with our assumption.
>
> By "cost of annotating," we refer to the time or financial resources required to annotate slices at a given annotation density and across a specific number of volumes. We believe annotation cost can be decomposed into two components: (1) the time needed to identify/localize the target and (2) the time needed to find the contour and draw or refine the mask. For component (2), the time required for annotating a batch of slices sampled in a nearly unbiased manner should be similar across batches (as the ratio of slice containing the target converges in probability to the ratio of slices containing target in the dataset). For component (1), at lower annotation densities—where selected slices are unlikely to capture the same object—the localization time should also be roughly proportional to the number of slices annotated. On the other hand, for easily distinguishable organ-like structures such as bones, (1) is expected to be minimal, as experienced annotators can identify the target almost instantly.
>
> Based on this rationale, we believe our assumption holds realistically in most scenarios.
>
> **Although choosing representative segmentation models per dataset is a strength of the manuscript, even the authors state that there might be an interaction between dataset, architecture and observations. Without some additional experiments (ablation studies), the question is whether the same observations hold for different architectures (as indicated in Section 4).**
>
> We acknowledge the reviewer's feedback. We clarified the need for further research on the interaction between datasets, architectures and observations in the Discussion and Conclusion section.
>
> **There is no table 4 (Section 3.2), authors probably meant figure 4?**
>
> We thank the reviewer for catching this mistake. We have made the correction.
>
> **Figure 4 ATLAS with M.I. stops at density = 0.4, are there no values for 0.8? [Why is ATLAs with M.I. for density 0.8 not available in Figure 4?]**
>
> We applied weighted sampling to ATLAS to achieve reasonable interpolation performance. Existing literature suggests that this weighting process can be done in an unsupervised manner, so we did not explore it further. At density = 0.8, all slices containing the ROI would be selected and manually annotated, leaving no opportunity for the Slice Interpolation algorithm to operate. Please check the newly added appendix under "Numerical results of Model Performance when Trained with Mask Interpolation" for details.
>
> **For the M.I. algorithms experiments (Figure 4), it would be interesting to see the results for density 1.0 (dense annotations) as comparison (upper limit).**
>
> In response to the reviewer's request, we added the corresponding data point at dense annotation to the figure.
>
> **Appendix C is not mentioned in the main manuscript**
>
> In response to the reviewer's suggestion, we now mention all appendices in the main text.
>
> **As various datasets are used and some data characteristics are discussed (e.g. in Section 3.2 FGT), it would help the reader to see some examples of the reference data and the predictions (in the appendix).[Representative qualitative examples of some experiments]**
>
> We agree, and we have added these visual examples to the appendix under "Visualization of selected targets" to improve clarity for the reader.
>
> **Section 4 could be renamed Discussion and Conclusion, as both are combined into one section**
>
> In response to the reviewer's suggestion, we have updated the section title accordingly.
>
> **There are some typos and missing spaces in the appendix.**
>
> We appreciate the reviewer's careful review and have corrected the typographical errors.

---

> > ### Comment · Reviewer_xr3K · 2025-03-11
> > **Official Comment**
> >
> > Thank you to the authors for the clarifications and revisions. All questions from the previous review comments have been addresses.

---

> ### Author Response · Authors · 2025-03-14
>
> Thank you for your quick turnaround and thoughtful feedback. We truly appreciate your time and effort in reviewing our work.

---

### Author Rebuttal · Authors · 2025-03-08

**Rebuttal:**

We sincerely thank the reviewers for their valuable feedback, which has significantly contributed to improving the presentation of our work. We have incorporated most of the suggested changes and have provided a point-to-point response to each reviewer’s comments.

Please find attached a revised version of the manuscript addressing the reviewers' requests and concerns. In addition, we have refined most of the figures to include error bars, axis grids, and higher resolution. We have also added several appendices to provide more detailed information regarding numerical results, rationales, and implementations.

**Supporting Material:**

/attachment/a93ee95a823c13057e405307433d37a14e12ede5.pdf

---

### Meta-Review · Area_Chair_CBcw · 2025-03-23

**Recommendation:** Accept (Poster)
**Confidence:** 5

**Metareview:**

Summary:
The paper is an empirical study of how to choose the best slices for annotation given a specified annotation budget. Key results are:
1. Annotate more volumes with fewer slices --> leads to improved performance.
2. Random slice selection may work better than unsupervised active learning.
3. Slice selection at fixed interval is sufficient as mask interpolation provided mixed results.

Consensus amongst the reviewers was that the paper has some empirical value and may benefit the research community, although rigorous statistical testing has not been done. The authors are encouraged to consider adding statistical testing to the paper.

Metareviewer decision:
Accept